# Quality and Satisfaction in Health Care: A Case Study of Two Public Hospitals in Trujillo, Peru

**DOI:** 10.3390/ijerph22071119

**Published:** 2025-07-16

**Authors:** Ariane Morales-Garrido, Brigitte Valderrama-Pazos, Jeremy García-Carranza, Alexis Horna-Velásquez, Willy Reyes-Anticona, Anlli Estela-Vargas, Walter Rojas-Villacorta

**Affiliations:** 1Escuela de Medicina, Facultad de Ciencias de la Salud, Universidad César Vallejo, Trujillo 13001, Peru; amoralesga26@ucvvirtual.edu.pe (A.M.-G.); brvalderramap@ucvvirtual.edu.pe (B.V.-P.); jgarciaca23@ucvvirtual.edu.pe (J.G.-C.); arhornav@ucvvirtual.edu.pe (A.H.-V.); wreyesan@ucvvirtual.edu.pe (W.R.-A.); aestelava01@ucvvirtual.edu.pe (A.E.-V.); 2Dirección de Investigación, Universidad César Vallejo, Trujillo 13001, Peru

**Keywords:** service quality, user satisfaction, responsiveness, security, public hospitals

## Abstract

(1) Background: The Peruvian healthcare system is widely regarded as deficient, with ongoing improvements identified as a key area of need. This study sought to assess user satisfaction and the quality of care in two public hospitals in Trujillo. (2) Methods: A non-experimental, cross-sectional, and correlational study was carried out. A group of 384 people who used two public hospitals in the city of Trujillo was studied. The people in the study were chosen based on the researchers’ convenience sampling. Information was collected using a survey based on the SERVQUAL model. This survey was used to evaluate the quality of service. Descriptive and inferential analyses were performed, including Spearman’s correlation and multinomial logistic regression to assess associations and identify key predictors of perceived service quality. (3) Results: The results indicated that 97.66% of the users perceived a low quality of care and 100% expressed dissatisfaction with the services. The most critical dimensions were reliability and responsiveness, while tangible items obtained better results. A positive correlation (rho = 0.723) was identified between quality of care and user satisfaction, with empathy (rho = 0.559) and safety (rho = 0.543) emerging as the most influential dimensions. (4) Conclusions: Responsiveness and Security were identified as key predictors of the perceived service quality in two public hospitals in Trujillo, Peru. Despite high empathy correlations, only timely care and safety significantly influenced satisfaction. The findings align with SDG 3, calling for improved efficiency and humanized care in public health services.

## 1. Introduction

The quality of healthcare is a matter of global concern due to its profound impact on human health. The World Health Organization (WHO) characterizes quality of care as a high level of competence that ensures the effective utilization of available resources while minimizing risks to the user. This approach aims to achieve a high level of satisfaction and a positive impact on health [1]. The first component is professional care, which involves the application of science and technology to address the health concerns of patients. The second component is interpersonal care, encompassing the social exchange between patients and physicians from cultural and economic perspectives [2].

The concept of user satisfaction represents a novel approach to healthcare, wherein the term “patient” is substituted with “user” or “customer”. This shift in language signifies a transformation in the paradigm of modern healthcare management, with an emphasis on enhancing the performance and efficiency of healthcare systems within the context of their business nature. The concept of healthcare quality is inherently challenging to objectively measure. Its evaluation is thus indirect, through several factors: economic, health impact (in terms of reduction of illnesses and deaths), and, finally, user satisfaction [3]. In recent decades, the analysis of user satisfaction has focused on their perception of the service they receive. This approach considers the user as a priority in care, recognizing that their experience is fundamental to the evaluation of medical care [4]. For this reason, excellence in healthcare services is fundamental to achieving patient approval. This perception of quality exerts a direct influence on the user experience and satisfaction [5].

User satisfaction is an essential component of health service evaluation because it encompasses not only the effective treatment of patients’ health problems, but also their experience during the care process [6]. Research has identified factors that influence user satisfaction, such as staff courtesy, responsiveness, the transparency of information provided, and the empathy demonstrated by healthcare professionals [6]. In addition, studies have shown that a lack of trained customer service staff and inadequate infrastructure contribute to a negative user experience [7]. From two theoretical perspectives, it is recognized that a satisfied user improves the perception of quality and that service quality promotes satisfaction. In addition, service quality has been linked to behavioral intention. Measuring user satisfaction is critical because it provides information about the structure, processes, and outcomes of care and influences user behavior. Empirical evidence confirms that basic medical services improve user trust and satisfaction [8].

Studies of quality of care in different regions provide important data. Aiken (2012) reported 80% patient satisfaction at Cedars-Sinai Medical Center in Los Angeles, focusing on nursing care and hospital cleanliness [9]. Friedel (2023) found high satisfaction with physician competence at the University Hospital Essen (Germany), although problems with administrative efficiency and long waiting times highlighted areas for improvement [10]. Similarly, Alabdullah’s research at the University Hospital of Vietnam found significant satisfaction with staff professionalism, but noted concerns about infrastructure and waiting times [11]. In Argentina, Araujo (2022) reported 80% overall satisfaction at the Alejandro Posadas National Hospital, where improvements were made in food services and patient experience, reflecting a commitment to improving hospital quality [12].

In Peru, several studies have shown a correlation between low user satisfaction and factors such as infrastructure, resource management, and quality of care [13]. Febres (2020) reported an overall satisfaction of 60% at the Daniel Alcides Carrión Hospital in Huancayo, although he identified gaps in accountability, reliability, and infrastructure; patients rated perceived safety and empathy in care positively [14]. Córdova (2021) found a significant correlation between patient satisfaction and perceived quality in terms of staff, facilities and efficiency in two hospitals in Lima [15]. However, a worrying trend of general dissatisfaction persists in Peruvian hospitals, as highlighted by Barrios (2021), who reported that 75% of patients in Lima expressed dissatisfaction with accessibility, particularly in primary care [16,17].

It is crucial to acknowledge that health professionals do not undergo customer service training during their university education, resulting in suboptimal user care. Consequently, there is a necessity to examine the factors that influence patient satisfaction in hospitals. By doing so, it becomes feasible to propose enhancements in care methodologies and public policies aimed at optimizing excellence in healthcare across all levels of the healthcare system. The cultivation of a patient-centered culture and the perpetual pursuit of excellence would be fostered by this approach [10,18]. Various methods have been used to improve services and user satisfaction. Among these studies, Peruzzo et al. [19] proposed the use of patient perspectives to guide improvements in medical care, prioritizing experiential factors. Another methodology to improve healthcare service quality, patient outcomes, and operational efficiency is Healthcare 4.0 technology, although it presents some challenges [20].

The central research question guiding this study is as follows: What is the user’s perception of the quality of service provided in two hospitals in the city of Trujillo? This question comes up because a Peruvian newspaper said that, in 2024, public hospitals in Trujillo were having problems. These problems included not having enough workers, having poor building conditions, and having bad management [21].

The main objective of this study is to evaluate the quality of care and user satisfaction in two public hospitals in Trujillo, Peru, in order to generate empirical evidence that supports the design of locally adapted improvement strategies. Beyond commonly reported issues such as dissatisfaction and infrastructure deficits, the study emphasizes less explored dimensions—such as staff empathy and perceptions of safety—that are essential for enhancing the user experience. By aligning with Sustainable Development Goal 3 (SDG-3), which promotes health and well-being for all, the study contributes to the development of more equitable and efficient healthcare systems, particularly for the retired population in Trujillo [22,23]. These findings may also provide a useful model for addressing similar challenges in other middle-income countries.

Based on the SERVQUAL model and the reviewed literature, it was proposed that perceived service quality is significantly associated with user satisfaction in public hospitals. Additionally, it was proposed that certain dimensions of the model, such as tangibles, reliability, responsiveness, security, and empathy, could predict users’ perception of the overall service quality. To this end, a multinomial logistic regression analysis was conducted to determine which dimensions best explain variations in user perception.

## 2. Materials and Methods

### 2.1. Type and Design of the Investigation

This study employs a cross-sectional, quantitative design to analyze the relationship between service quality and user satisfaction in two public hospitals in Trujillo. This design allowed us to capture users’ perceptions at a given moment in order to contribute to the formulation of improvement strategies adapted to the hospital context.

### 2.2. Variables

The independent variable in this research is the quality of service provided in two public hospitals in the city of Trujillo. The dependent variable is user satisfaction with the service received in these same hospitals. Furthermore, sociodemographic variables such as age, gender, and income level were collected, as these have been identified as possible confounding variables in the relationship between perceived quality and user satisfaction.

### 2.3. Population, Sample, and Sampling

The population was studied in two public hospitals in the city of Trujillo (Peru): Hospital Belén de Trujillo and Hospital Regional Docente de Trujillo (HRDT). Both are level III hospitals that provide comprehensive and specialized care to the population. Individuals over the age of 18 who had utilized the services of one of the two hospitals were considered for participation, with the objective of collecting relevant information regarding their experiences in the utilization of health services. Individuals under the age of 18 and those who had not utilized hospital services in the previous year were excluded from the study. This means the responses are based on actual experiences.

Individuals who were only temporary users or tourists were also excluded, as their experiences in Trujillo hospitals do not represent the perspective of permanent residents.

A non-probabilistic convenience sampling technique was employed, as the population is infinite and this approach is the most feasible and expeditious way to access the population [24]. The sample size was calculated at 384 individuals, with a confidence interval (CI) of 95%, implying a z value of 1.96, an error (e) of 5%, and an estimated proportion of the population (p) of 0.5. The following formula was used:n = z^2^ × p × (1 − p)/e^2^(1)

### 2.4. Data Collection Techniques and Instruments

For the data collection in this descriptive cross-sectional research, the survey was used as the main technique due to its effectiveness in obtaining detailed information about users’ perceptions in less time. The data collection period was approximately one month in March 2025. The survey was administered face-to-face to users who were being attended for various health problems in the two hospitals.

The instruments developed by Vigo (2020) [25] were used to assess user satisfaction (based on a SERVQUAL model) and service quality. Both instruments consisted of questionnaires with 25 and 26 questions, respectively. The first questionnaire consisted of 25 questions and covered five basic dimensions: Tangible Elements, Reliability, Responsiveness, Security and Empathy. Each dimension is detailed by questions that attempt to capture the user experience, from the comfort and modernity of the facilities to the attitude and willingness of the staff to solve problems. The second user satisfaction questionnaire consisted of 26 questions and consisted of three dimensions: Perceived Performance, Expectations, and Satisfaction. The questions cover aspects such as the user’s perception of accessibility and the reasonableness of prices, the laboratory’s fulfillment of promises, and the friendliness of the staff. It also assesses whether the services provided meet the client’s expectations and inspire confidence. Both questionnaires have Crombach’s alpha values of 0.893 and 0.870, respectively, which ensures their measurement accuracy. Both also have a response scale from never to always.

### 2.5. Statistics Analysis

The analysis of the data was conducted through the utilization of both descriptive and inferential statistics. In the initial case, general characteristics, quality levels, and user satisfaction were distributed in frequency tables and percentages. The survey responses were previously coded by assigning a numerical value. Subsequently, a Microsoft Excel spreadsheet was utilized to calculate the total scores for quality of service and user satisfaction. These scores were then analyzed to determine their correlation. Prior to this, IBM SPSS version 27 statistical software was used to analyze the distribution of normality for both variables by means of the Kolmogórov–Smirnov test, where a non-normal distribution was obtained (*p* > 0.05). Consequently, Spearman’s correlation test was employed to assess the relationship between the variables of interest. To further analyze the prediction of overall service quality perception, multinomial logistic regression was used, with the likelihood ratio test as the comparison criterion.

### 2.6. Ethical Aspectsb

The ethical framework of the Universidad César Vallejo (UCV) was assessed in this study, as outlined in the institution’s Code of Ethics (Rectoral Resolution N° 760-2007/UCV [26]. This code emphasizes the values of integrity, intellectual honesty and objectivity, and impartiality in professional interactions. It also underscores the autonomy of participants in determining their involvement. Additionally, Law N° 26842 stipulates the right to healthcare within the framework of the law, adhering to the ethical principles of respect, honesty, and truthfulness [27]. The research was reviewed and approved by the Research Ethics Committee of the UCV Research Department on 18 February 2025.

## 3. Results

Table 1 presents the primary characteristics of the 384 respondents. The data indicate that the majority of the participants are between 40 and 59 years of age, constituting 56.25% of the sample. With respect to gender, 61.20% of the participants are female and 38.80% are male. In terms of income, the majority of participants (40.10%) earn less than PEN 1000, which is less than the minimum wage in Peru.

Table 2 shows the quality of care as perceived by users. The results show that the vast majority (n = 375; 97.66%) consider the quality of service to be low. Only a small percentage, 2.34%, perceived the quality of care to be average.

Table 3 shows the distribution of frequencies and percentages in five dimensions of quality of service: tangible elements, reliability, responsiveness, safety and empathy, classified into high, medium and low levels. The tangible elements have a predominantly favorable perception, at 40.10%, while reliability stands out as an area of opportunity at the low level, with 42.97%. Similarly, responsiveness has a predominantly medium level (88.54%), but with room for improvement in speed. Security and empathy show similar trends, with a majority at the medium level, with 53.38% and 85.15%, respectively; however, both require attention to strengthen trust and emotional connection.

Table 4 shows the data on user satisfaction with the hospital service. The data show that all participants rated their satisfaction as low (n = 384, 100%), with no medium or high levels reported (0%). This result indicates a critical deficiency in the quality of service received.

The results in Table 5 demonstrate a significant relationship between service quality and user satisfaction in two hospitals (*p* < 0.05). This relationship is positive and strong, with a Spearman’s coefficient (ρ) of 0.723, indicating that higher service quality is associated with higher user satisfaction. All dimensions of service quality demonstrate a positive and statistically significant correlation with user satisfaction. Empathy (rho = 0.559) demonstrates the strongest correlation, indicating that staff’s capacity to comprehend and address individual user needs significantly contributes to their positive perception of the service. Safety (rho = 0.543) exhibits a moderate-to-high relationship with satisfaction, underscoring the significance of safety perceptions in healthcare services. Reliability, with a correlation of 0.522, is another salient factor in user perception, underscoring the significance of the service’s ability to deliver on its commitments. Responsiveness, with a moderate correlation of 0.435, demonstrates that speed and responsiveness influence user satisfaction, though to a lesser extent than other dimensions. Finally, tangible elements have a rho of 0.405, indicating that the physical aspects of the service have a positive, albeit small, impact on user satisfaction.

Table 6 shows that the multinomial regression indicates that the dimensions of responsiveness and safety are significant predictors of the level of perceived quality, even when controlling for the other variables in the model. This supports the findings of the bivariate analysis and highlights their importance in improving hospitals.

## 4. Discussion

The data in Table 1 shows that the majority of users are women (61.20%), followed by men (38.80%). The majority of users are between 40 and 59 years old (56.25%), with a proportion under 40 years old (32.55%) and over 60 years old (11.20%). This is in line with the findings of Febres-Ramos and Mercado-Rey (2020), who also observed a predominance of women (61.0%) over men (39.0%) and reported similar distributions by age group, with the highest percentage in the 36–45 age group (31.0%) [14]. While this similarity strengthens the findings, it also suggests the need to explore the underlying factors, such as gender roles, that condition the use of health services. The higher representation of women may be related to their greater awareness of preventive health measures [28]. In addition, most respondents reported an income below the Peruvian minimum wage, which, although not a definitive determinant of health care quality, could influence patient outcomes, as noted by Soto-Becerra et al. [29]. This finding raises critical questions about how economic inequalities affect perceptions of and access to care, and highlights a structural dimension that should be addressed in future studies.

An analysis of the quality of service reveals that the majority of users perceive it as “low”, with 97.66% of the participants indicating as such (see Table 2). This finding is indicative of the inefficiency of the health services. This perception of low quality can be attributed primarily to deficiencies in infrastructure, inefficient administrative processes, and a shortage of trained personnel. The analysis further reveals that a lack of empathy in care and elevated expectations derived from patients’ socioeconomic profiles also contribute to this perception. These findings underscore the need for comprehensive improvements in resources, management, and training to enhance the user experience. A mere 2.34% of participants regarded it as “average”. This finding aligns with the observations of other researchers who have documented the substandard quality of care in Peruvian public hospitals. Córdova-Buiza et al. [15] noted that, in 2019, MINSA hospitals were considered substandard. Carhuancho et al. [30] reported that 74.6% of patients in level III hospitals in Lima rated their quality as “regular” due to problems such as disorganization and long waiting times. However, Hernández-Vásquez et al. [31], based on the ENAHO 2018, found that 74.3% of users rated the service received as good or very good.

As illustrated in Table 3, the tangible elements (i.e., medical equipment and facilities) were perceived to be of a medium (51.83%) and high (40.10%) level. Reliability was rated low (55.73%), indicating a need for improvement in this area. Responsiveness (88.54%), safety (53.38%), and empathy (85.15%) were predominantly at medium levels, suggesting the need to enhance trust and emotional connection with patients. The importance of tangible aspects related to user satisfaction and customer loyalty is well-documented [32]. The results concerning the dimensions of the independent variable (reliability, responsiveness, safety, empathy, and tangible aspects) are consistent with those reported by Arce Huamani and Aliaga-Gastelumendi [33]. A similar observation was made in the study by De la Cerna-Luna [34], which found that the dimensions of safety, tangible aspects, and empathy influenced the level of satisfaction of users of the Edgardo Rebagliati Martins National Hospital in Lima.

The results show that 100% of respondents in public hospitals rated their satisfaction as low (see Table 4), reflecting a critical situation in the quality of services provided, in line with previous research in Peruvian public hospitals [31,33,35]. In contrast, a study in a hospital in Chachapoyas reported that 57% of users were satisfied, while 43% expressed moderate satisfaction [36]. Low satisfaction in Peruvian public hospitals is related to the lack of sufficient resources, especially in areas of extreme poverty, which leads to poor care and the overloading of larger hospitals with basic services that should be provided in primary care [15]. Other international studies confirm this problem. In Argentina, 80% satisfaction was reported due to the empathy of the staff, although shortcomings in the quality of the meals were noted [12]. In Germany, Zarei found that 75% of patients rated medical competence positively, but criticized waiting times and administrative procedures [18]. These factors are consistent with the problems identified in the present study, such as administrative inefficiency. Finally, it is crucial to improve the quality of Peruvian hospital services through strategies focused on humanized care and resource optimization, as suggested by Cordova-Buiza et al. [15].

The inferential analysis (Table 5) of the evaluated hospitals showed a positive and significant correlation between service quality and user satisfaction (rho = 0.723; *p* < 0.05), highlighting the urgency of improving the level of care. Among the dimensions evaluated, empathy (rho = 0.559) stood out as the most influential in user satisfaction, highlighting the importance of humanized treatment according to Rodriguez [33,37]. Safety showed a moderate correlation (rho = 0.53), relevant due to its impact on patient confidence [38]. In addition, reliability (rho = 0.522) and responsiveness (rho = 0.435) were identified as key factors for overall user perception. These results are consistent with previous research in Peruvian hospitals that indicates how structural and operational limitations negatively affect satisfaction [15,31,37]. On the other hand, international studies such as that of Zarei [18] found similar problems in hospitals in Germany, where waiting times and inefficient administration reduced satisfaction despite the high qualifications of the medical staff. Finally, the importance of implementing policies that optimize resources, improve safety standards, train staff emotionally, and modernize infrastructure to increase the quality perceived by users is underscored.

The multinomial logistic regression analysis identified which dimensions of the SERVQUAL model significantly influence users’ overall perception of service quality. According to the results presented in Table 6, the dimensions of responsiveness (D3) (χ^2^ = 16.436, *p* = 0.000) and Security (D4) (χ^2^ = 6.829, *p* = 0.033) emerged as significant predictors of perceived service quality, even when controlling for the other SERVQUAL dimensions. These findings are consistent with prior studies suggesting that prompt and efficient responsiveness can reduce patient wait times, thereby maintaining high levels of patient satisfaction [39,40]. Similarly, security in hospital settings fosters trust and a sense of safety among users, contributing to the delivery of high-quality medical care [41,42]. Although tangibles (D1) and reliability (D2) were not significant predictors, users appear to place greater value on the effectiveness of care over physical resources. This aligns with previous research indicating that certain service dimensions may exert a stronger influence on user satisfaction than others [40,43,44]. Moreover, reliability may be more readily perceived by healthcare professionals than by users themselves [45,46]. Finally, while empathy (D5) showed a strong bivariate correlation with satisfaction, it was not statistically significant in the multivariate model. This underscores the importance of prioritizing timely care, perceived safety, and humanized service delivery in hospital settings.

The findings reveal significant opportunities for improving health service delivery and underscore the need to implement policies and allocate sufficient resources to public hospitals. These policies should increase competitiveness and encourage preference for public over private health services. However, the study has important limitations, such as the use of non-probability sampling, which limits the generalizability of the results to the general population. In addition, the reliance on self-reported data introduces the possibility of bias, as responses may be influenced by individual perceptions and experiences rather than objective measures. Finally, resource-constrained settings face unique challenges, such as inadequate infrastructure, scarce staff, and limited financial resources. This may limit the ability of institutions to meet user expectations on several of the dimensions assessed by SERVQUAL. Future research should address these issues by using probability sampling methods and incorporating objective data collection techniques to ensure more reliable and representative results.

## 5. Conclusions

This study revealed a critical gap in service quality and user satisfaction across two public hospitals in Trujillo, Peru, with 100% of users reporting dissatisfaction. Multinomial logistic regression identified responsiveness and security as the most significant predictors of perceived service quality, highlighting the essential role of prompt attention and perceived safety in shaping user perceptions. While empathy demonstrated a strong bivariate correlation, it lost significance in multivariate analysis, suggesting that its effect may be mediated through other service dimensions. These findings underscore the urgent need for targeted policies that strengthen operational efficiency, optimize human resources, and enhance the humanization of care. Aligning with Sustainable Development Goal 3 (Good Health and Well-being), this study emphasizes the importance of equitable, responsive, and patient-centered healthcare services to improve quality and satisfaction, especially in resource-limited settings.

## Figures and Tables

**Table 1 ijerph-22-01119-t001:** Characteristics of users of two public hospitals in the city of Trujillo.

	Frequency (n)	Percentage (%)
Age		
<40 years	125	32.55
40–59 years	216	56.25
≥60 years	43	11.20
Gender		
Male	149	38.80
Female	235	61.20
Salary Income		
<1000	192	50.00
1000–2500	154	40.10
2500–3500	38	9.90
>3500 soles	0	0
Total	384	100.00

**Table 2 ijerph-22-01119-t002:** Quality of service perceived by users of two public hospitals in the city of Trujillo.

Level	Frequency (n)	Percentage (%)
High	0	0%
Medium	9	2.34%
Low	375	97.66%
Total	384	100.00%

**Table 3 ijerph-22-01119-t003:** Frequency distribution and percentages of independent variable dimensions.

Dimensions of Quality of Service	Frequency (n (%))			
	High	Medium	Low	Total
Tangible elements	154 (40.10)	199 (51.83)	31 (8.07)	384 (100.00)
Reliability	12 (3.13)	158 (41.14)	214 (55.73)	
Responsiveness	14 (3.65)	340 (88.54)	30 (7.81)	
Security	14 (3.65)	205 (53.38)	165 (42.97)	
Empathy	14 (3.65)	327 (85.15)	43 (11.2)	

**Table 4 ijerph-22-01119-t004:** User satisfaction in two public hospitals in the city of Trujillo.

Level	Frequency (n)	Percentage (%)
High	0	0
Medium	0	0
Low	384	100
Total	384	100.00

**Table 5 ijerph-22-01119-t005:** Spearman correlation of the independent variable (quality of service) with the dependent variable (user satisfaction).

Variable	User Satisfaction		
	rho(ρ)	*p*-Value	n
Quality of service	0.723 ^a,b^	0	384
D1: Tangible elements	0.405	0	384
D2: Reliability	0.522	0	384
D3: Responsiveness	0.435	0	384
D4: Security	0.543	0	384
D5: Empathy	0.559	0	384

^a^ Correlation is significant at the 0.05 level (bilateral). D1, D2, D3, D4, and D5: Quality of service Dimensions. ^b^ R^2^ = 52.3% (Determination coefficient).

**Table 6 ijerph-22-01119-t006:** Tests of the multinomial regression model using the likelihood ratio method.

Dimension (SERVQUAL)	Chi-Square (χ^2^)	gl	*p*-Value ^a^	Significance
D1: Tangibles	0.504	2	0.777	Not significant
D2: Reliability	0.400	2	0.819	Not significant
D3: Responsiveness	16.436	2	0.000	Significant
D4: Safety	6.829	2	0.033	Significant
D5: Empathy	4.304	2	0.116	Marginal (*p* ≈ 0.1)

^a^ Correlation is significant at the 0.05 level (bilateral). D1, D2, D3, D4, and D5: Quality of service Dimensions.

## Data Availability

All data generated or analyzed during this study are included in this published article.

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
