# Peer review of "Quality and Satisfaction in Health Care: A Case Study of Two Public Hospitals in Trujillo, Peru"

_ijerph, 2025, doi:10.3390/ijerph22071119_

Round 1

Reviewer 1 Report

Comments and Suggestions for Authors

You should construct hypothesis (ses) you try to test. Literature review should be followed by construction of hypotheses. This suggested approach will giv the readers clear directions of your research.

Literature review could be further extended as there are many published articles covering the quality of care.

Author Response

1.You should construct hypothesis (see) you try to test. Literature review should be followed by construction of hypothesis. This suggested approach will give the readers clear directions of your research.

Response 1: Thank you for your observation. The hypothesis was added at the end of introduction (line 112-117).

2. Literature review could be further extended as there are many published articles covering the quality of care.

Response 2: We appreciate your suggestion to expand the literature review. However, we believe that the current section already incorporates relevant and updated studies that adequately support the paper's conceptual framework, particularly with regard to the SERVQUAL model and the perception of quality in public hospital settings. Therefore, we have decided to keep the revision as it is, ensuring the manuscript's coherence and consistency. However, we are open to considering any specific recommendations for additional studies.

Reviewer 2 Report

Comments and Suggestions for Authors

Thank you for the opportunity given to me to review this paper. Reviewer had several queries regarding this paper before could be accepted.

  1. In line 119-24, Authors need to simplify their study design as follow: cross-sectional study design. There is no need to mention correlation and non-experimental.
  2. in line 113-117, change hypotesis into objective or aim.
  3. In variables section, please specify what if the confouding variables collected in this study?
  4. In line 138, authors wrote "those who had not utilized hospital services in the previous year were excluded". please explain about this sentence, because it means if there is a participant who utlized health service during data collection but not utilizing in the last year, will it be excluded? if yes, explain it.
  5. Authors decided to use non-probabilistic. This introduce bias to this study because an analytical study is preferred to use probability sampling. Please explain why use non-probabilistic and explain how authors control the bias.
  6. Please explain how authors analyze the data in the method section.
  7. How authors divide satisfaction score into four groups? please explain in the method.
  8. Please re-analyze the data using multivariate analysis tehcnique such as ordinal or multinomial logistic to control confouding. Spearman rank test is not preferred anymore as it can't control confounding. 

Author Response

1. In line 119-24, Authors need to simplify their study design as follow: cross-sectional study design. There is no need to mention correlation and non-experimental.

 Response 1: It was corrected.

2. in line 113-117, change hypotesis into objective or aim.

Response 2: ok. It rewritten (line 103-114).

3. In variables section, please specify what if the confouding variables collected in this study?

Response 3: It was added (line 125-127).

4. In line 138, authors wrote "those who had not utilized hospital services in the previous year were excluded". please explain about this sentence, because it means if there is a participant who utlized health service during data collection but not utilizing in the last year, will it be excluded? if yes, explain it.

Response 4: Thank you for your comment. To ensure that responses are based on actual and complete experiences during the defined analysis period, those who did not use hospital services in the previous year were excluded. This ensures the comparability and validity of the data. This was explained on line 136.

5. Authors decided to use non-probabilistic. This introduce bias to this study because an analytical study is preferred to use probability sampling. Please explain why use non-probabilistic and explain how authors control the bias.

Response 5: We used the probabilistic method because it reduces sampling time and provides access to the population more quickly. However, we understand that this method may introduce bias. Fortunately, the high reliability of the instruments helps reduce response bias.

6. Please explain how authors analyze the data in the method section.

Response 6: It was added (Section 2.5)

7. How authors divide satisfaction score into four groups? please explain in the method.

Response 7: We appreciate your comment. Regarding your comment, we would like to clarify that the satisfaction score was not divided into four groups. Rather, a classification into three levels—high, medium, and low—was used. This categorization was applied to evaluate user satisfaction and service quality following the methodology established in the instruments adapted from the SERVQUAL model.

8. Please re-analyze the data using multivariate analysis tehcnique such as ordinal or multinomial logistic to control confouding. Spearman rank test is not preferred anymore as it can't control confounding. 

Response 8: Thank you for your observation. In addition to Spearman's test, a multinomial regression analysis was performed. The results of the analysis were placed in Table 6, thus reinforcing the bivariate analyses of Spearman's correlation and strengthening the research. This was added to the discussion regarding Table 6 (lines 319–336).

Reviewer 3 Report

Comments and Suggestions for Authors

Quaility and satisfaction in healthcare are important topic worldwide. You paper well describes quality dimensions that influence satisfaction the most. I think a serious defficiency of your paper is sample of patients that are 100% disatisfied with the service and 97% find quality of the health service low. In terms of statistical analysis sample should be bigger including patients that experienced better quality of health service and are more satisfied - that probably means one more health (probably private) facility. Better data could give you valuable insight into factors, that contributed to better satisfaction

Author Response

Coments 1: Quaility and satisfaction in healthcare are important topic worldwide. You paper well describes quality dimensions that influence satisfaction the most. I think a serious defficiency of your paper is sample of patients that are 100% disatisfied with the service and 97% find quality of the health service low. In terms of statistical analysis sample should be bigger including patients that experienced better quality of health service and are more satisfied - that probably means one more health (probably private) facility. Better data could give you valuable insight into factors, that contributed to better satisfaction.

Response 1: We recognize that 100% and 97% may be extreme values; however, they are consistent with user reports and complaints (Diario Correo, 2025: Available at: https://diariocorreo.pe/edicion/la-libertad/hospitales-de-trujillo-afrontan-crisis-cronica-la-libertad-peru-noticia/?ref=dcr). These reports refer to structural problems in public hospitals in Trujillo. On the other hand, the high reliability of the SERVQUAL survey (α = 0.893) allows for greater reliability of the responses. Thus, rather than being a limitation, these findings are evidence of the accumulated discomfort of users in two Level III public hospitals that share the chronic problems of the Peruvian health system. For future studies, we agree that it is necessary to incorporate institutions with higher satisfaction indices. This will allow us to identify differentiating factors and good practices that can be replicated in similar settings.

Round 2

Reviewer 1 Report

Comments and Suggestions for Authors

Well done. You used term "assurance" in many places. I am not quite sure the meaning of "assurance." In healthcare research, "assurance" usually refers to "promise". "guarantee", etc. Is what you have in mind? If not, you may consider exploring some other word to reflect your intention.

Author Response

Comments: Well done. You used term "assurance" in many places. I am not quite sure the meaning of "assurance." In healthcare research, "assurance" usually refers to "promise". "guarantee", etc. Is what you have in mind? If not, you may consider exploring some other word to reflect your intention.

Response: Thank you for letting me know. It was a mistake in the translation. The correct term is "Security". It has been changed.

Reviewer 2 Report

Comments and Suggestions for Authors

Authors had revised according to the reviewers' suggestions. 

Author Response

We apreciate your feedback.

Reviewer 3 Report

Comments and Suggestions for Authors

Quality of health services and patiens' satisfaction are important topic worldwide. Both topics as well as statistical analysis  are well presented. Although low quality of health services and patients' disatisfaction are expected in Peruvian context, article lacks the abition for a bigger scope either in Peru or international comparison .

Author Response

Thank you for your thoughtful observation. While the primary objective of the study was to generate locally grounded evidence on healthcare service quality and user satisfaction in two public hospitals in Trujillo, we acknowledge the importance of situating these findings within a broader national and international context. The manuscript includes references to studies from countries such as the United States, Germany, Vietnam, and Argentina, as well as comparative research from various regions of Peru. However, we recognize that these were narrative contrasts rather than structured or analytical comparisons.
In future research, we intend to incorporate multicentric data and formally compare healthcare service indicators across regions or similar international settings. This will enhance external validity and allow for more robust benchmarking of user satisfaction trends across health systems. We appreciate your comment as a valuable suggestion to expand the scope and applicability of our findings